# Musculoskeletal simulation of limb movement biomechanics in Drosophila melanogaster

**Pembe Gizem Özdil**[*]
EPFL
Kempner Institute, Harvard University
pgozdil@fas.harvard.edu

**Chuanfang Ning**[*]
EPFL
chuanfang.ning@epfl.ch

**Jasper S. Phelps**     **Sibo Wang-Chen**     **Guy Elisha**     **Auke Ijspeert**     **Pavan Ramdya**
EPFL                     EPFL                    EPFL             EPFL                  EPFL

## Abstract

Computational models are critical for advancing our understanding of how neural, biomechanical, and physical systems interact to orchestrate animal behavior. Despite the availability of near-complete reconstructions of the *Drosophila melanogaster* central nervous system, musculature, and exoskeleton, anatomically and physically grounded models of fly leg muscles are still missing. Such models provide an indispensable bridge between motor neuron activity and joint movements. Here, we introduce the first 3D, data-driven musculoskeletal model of *Drosophila* legs, implemented in both OpenSim and MuJoCo simulation environments. Our model incorporates a Hill-type muscle representation based on high-resolution X-ray scans from multiple fixed specimens. We present a pipeline for constructing muscle models from morphological imaging data and for optimizing unknown muscle parameters specific to the fly. We then combine our musculoskeletal models with detailed 3D pose estimation data from behaving flies to achieve muscle-actuated behavioral replay in OpenSim. Simulations of muscle activity across diverse walking and grooming behaviors predict coordinated muscle synergies that can be tested experimentally. Furthermore, by training imitation learning policies in MuJoCo, we examine the effect of different passive joint properties on learning speed and find that damping and stiffness facilitate learning. Overall, our model enables the investigation of motor control in an experimentally tractable model organism, providing insight into how biomechanics contributes to the generation of complex limb movements. Moreover, it can be used to control embodied artificial agents to generate naturalistic and compliant locomotion in simulated environments.

## 1 Introduction

Understanding how to coordinate multiple limbs with many degrees of freedom (DoFs) to accomplish diverse motor tasks is a long-standing challenge in both motor control neuroscience and robotics. Performing such systems-level investigations hinges upon detailed knowledge of musculoskeletal structures and their mechanical properties. Neuromechanical models, therefore, serve as a crucial tool enabling rapid experimentation and hypothesis testing under controlled conditions (Edwards, 2010; Ausborn et al., 2021; Kim et al., 2022; Delp et al., 2007; Ramdya and Ijspeert, 2023).

Muscle-driven systems are characterized by intrinsic compliance and redundancy, which can simplify the control problems faced by artificial agents, as they do for animals in the real world (Geijtenbeek et al., 2013; Lee et al., 2018a; Wang et al., 2012). However, many existing models abstract away key elements of musculature and passive biomechanics (Aldarondo et al., 2024; Wang-Chen

---

[*]These authors contributed equally.

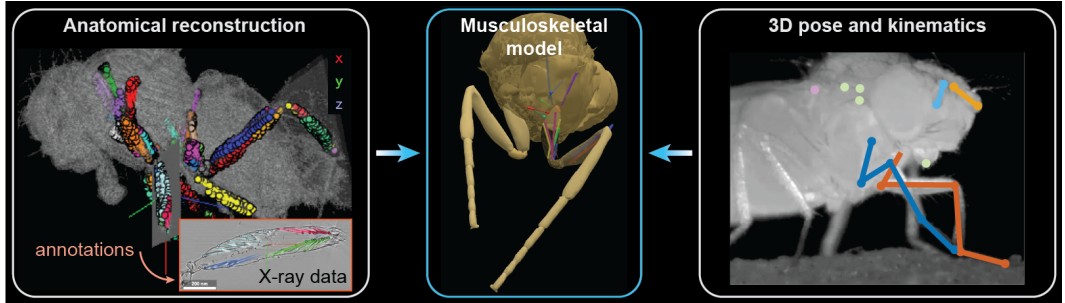

Figure 1: **Pipeline to develop *Drosophila* leg musculoskeletal models.** (Left) Anatomical data from multiple flies were used to determine muscle attachment points and fiber paths, providing constraints for constructing the muscle model in OpenSim. (Right) 3D pose estimation data from behaving animals were then used to train the muscle-actuated agent to recapitulate detailed kinematics in OpenSim and MuJoCo. (Middle) The resulting muscle model is informed by both leg anatomy and joint kinematics.

et al., 2024), limiting their ability to capture the richness of neuromuscular coordination. This, in turn, limits their utility for understanding how the nervous system generates robust, adaptive behaviors. Overcoming this gap will require modeling a system that generates sufficiently complex limb-dependent movements using a tractably small neuromuscular controller.

The fruit fly, *Drosophila melanogaster*, represents an ideal organism for this reverse-engineering challenge. Flies generate complex behaviors using a compact, well-characterized, and genetically accessible nervous system (Bellen et al., 2010; Rubin and Spradling, 1982). Despite substantial progress in anatomical (e.g., connectomic (Dorkenwald et al., 2024; Phelps et al., 2021)), neural (Aimon et al., 2019; Simpson, 2024; Chen et al., 2018), and behavioral characterization (Aranha and Vasconcelos, 2018; Özdil et al., 2024), recently published biomechanical models of *Drosophila* legs (Vaxenburg et al., 2025; Wang-Chen et al., 2024; Lobato-Rios et al., 2022) lack the anatomical and physiological accuracy required to model and investigate how motor networks coordinate muscles, limb kinematics, and behavior.

Here, we present the first biologically detailed musculoskeletal model of *Drosophila* legs across two widely used physics engines, OpenSim and MuJoCo. Our model extends NeuroMechFly (Wang-Chen et al., 2024; Lobato-Rios et al., 2022), a whole-body biomechanical simulation of the adult fly, by incorporating anatomically detailed muscle representations and physiological properties. Specifically, we implement a Hill-type muscle model informed by anatomical data (Azevedo et al., 2024; Lesser et al., 2024; Dinges et al., 2021) to actuate seven DoFs in three leg joints. Using this model, we predict the relative contributions of individual muscles to joint movements generated by real behaving flies. Furthermore, we study the influence of passive joint properties on the successful imitation of animal behavior. Our work integrates anatomical, physiological, and behavioral data into a unified modeling framework, providing a critical foundation for future investigations of *Drosophila* neuromuscular coordination and behavioral control. This modeling framework (Figure 1) can also accelerate the development of musculoskeletal models for other species, inform machine learning-based algorithms for embodied agents, and facilitate the control of bioinspired robots.

## 2 RELATED WORK

### 2.1 MUSCULOSKELETAL MODELS OF ANIMALS

Musculoskeletal models have emerged as powerful tools for investigating animal biomechanics. Open-source simulation platforms such as OpenSim (Delp et al., 2007), MyoSuite (Caggiano et al., 2022), and DART (Lee et al., 2018b), as well as commercial frameworks like HyFyDy (Geijtenbeek, 2021), have made it possible to test state-of-the-art learning algorithms on complex, high-dimensional systems. Use cases include the control of human arms for manipulation and legs for locomotion (Song et al., 2021; Caggiano et al., 2024). Although primarily developed to simulate human bodies (Sylvester et al., 2021; Seth et al., 2018), these tools have enabled the creation of

musculoskeletal models for a wide range of species, including rodents (Charles et al., 2018; Tata Ramalingasetty et al., 2021; Gilmer et al., 2025; DeWolf et al., 2024), primates (Ogihara et al., 2009; Chan and Moran, 2006), horses (van Bijlert et al., 2024), and birds like ostriches (Rankin et al., 2016; La Barbera et al., 2021). Researchers have also developed custom tools to study the interplay between neural circuits and biomechanics in worms (Zhao et al., 2024), insects (Guo et al., 2018), and other animals (Arreguit et al., 2023).

Recently, the fruit fly, *Drosophila melanogaster*, has become an increasingly prominent animal model for musculoskeletal simulations. The first anatomically realistic fly body model, NeuroMech-Fly, used micro-computed tomography (micro-CT) scans to construct a morphologically accurate 3D fly simulation. In its earliest incarnation, simplified antagonistic spring-damper muscles were used in the legs of NeuroMechFly to simulate locomotor dynamics (Lobato-Rios et al., 2022). More recent efforts in this (Wang-Chen et al., 2024) and other (Vaxenburg et al., 2025) fly body models have enhanced simulations by adding, for example, multimodal sensorimotor transformations to accomplish a wider range of behavioral tasks. However, these models still rely on abstract —position- or torque-based— joint controllers and lack realistic muscle actuation. We address this gap in biological fidelity in our work by explicitly modeling individual fly leg muscles based on detailed anatomical data.

## 2.2 MOTOR CONTROL AND MUSCLE SYNERGIES

Coordinating the activities of many muscles is a fundamental challenge in motor control. It has been proposed that the nervous system simplifies this complex problem by using muscle synergies—groups of muscles that are co-active, acting as functional units (Bizzi and Cheung, 2013; Ting and McKay, 2007). By combining a limited number of these synergies, the brain can more efficiently produce diverse movements without needing to control each muscle independently (Bizzi and Cheung, 2013; Saito et al., 2018). Muscle synergies have been identified and analyzed in real experimental data (Tresch et al., 2006; Sponberg et al., 2015; Al Borno et al., 2020), and inspired the development of controllers that operate within reduced-dimension muscle activation spaces (Berg et al., 2024). These synergy-based approaches facilitate robust coordinated movements while significantly reducing control complexity. For instance, human walking can be performed across a range of speeds and directions using just a few lower-limb synergies (Saito et al., 2018).

Muscle synergies have also been studied extensively in insects with respect to how sensory feedback reinforces coordinated muscle activation (Bidaye et al., 2018; Zill et al., 2018). By leveraging the complete wiring diagram of the *Drosophila* nervous system—known as the *connectome*—recent studies have identified specific neural circuits that may coordinate fly limb movements (Syed et al., 2026; Lesser et al., 2024). Nevertheless, because of the anatomical and functional complexity of *Drosophila* leg muscles, we cannot predict muscle activity patterns for a given behavior based on neural connectivity or neural recordings alone. In this work, we address this gap by (i) developing an anatomically detailed muscle model of *Drosophila* legs and (ii) simulating this model to reproduce measured joint kinematics across behaviors, allowing us to predict underlying active muscle synergy groupings.

## 2.3 DROSOPHILA LEG MUSCULATURE

Each fly leg is a multi-jointed appendage with at least seven DoFs across five joints (Lobato-Rios et al., 2022; Haustein et al., 2024). These joints are actuated by approximately 19 muscles, which in turn are controlled by approximately 69 motor neurons (Azevedo et al., 2024; Lesser et al., 2024; Soler et al., 2004). Notably, the muscle structure across legs is nearly identical, with a few exceptions including tergal depressor of the trochanter muscles in the middle legs that facilitate jump escape (Swank, 2012). Leg biomechanics are further complicated by biarticular muscles that span multiple joints to coordinate movements across body segments (Lesser et al., 2024). Additionally, the passive properties of joints, including their elasticity and damping, can either resist, or assist motion depending on the direction of movement (Ache and Matheson, 2013).

## 3 METHODS

### 3.1 ACQUISITION OF ANATOMICAL DATA

To reconstruct the front leg muscles, we used two publicly available datasets (Kuan et al., 2020; Dinges et al., 2021) and one custom dataset collected for this study. The custom dataset was acquired using synchrotron radiation $\mu$CT. Each of the three datasets captured different foreleg postures, allowing us to better understand how muscle attachment points vary across joint configurations. We focused on annotating muscle fibers that span the thorax–coxa, coxa–trochanter, and femur–tibia joints (Figure 2A) and grouped them by function as described in (Azevedo et al., 2024; Soler et al., 2004). For muscles located in the thorax, we used data from (Dinges et al., 2021); for the remaining leg muscles, we relied on (Kuan et al., 2020). Muscle attachment points were then cross-validated using our custom scan. Together, these datasets provided the anatomical basis for our muscle model.

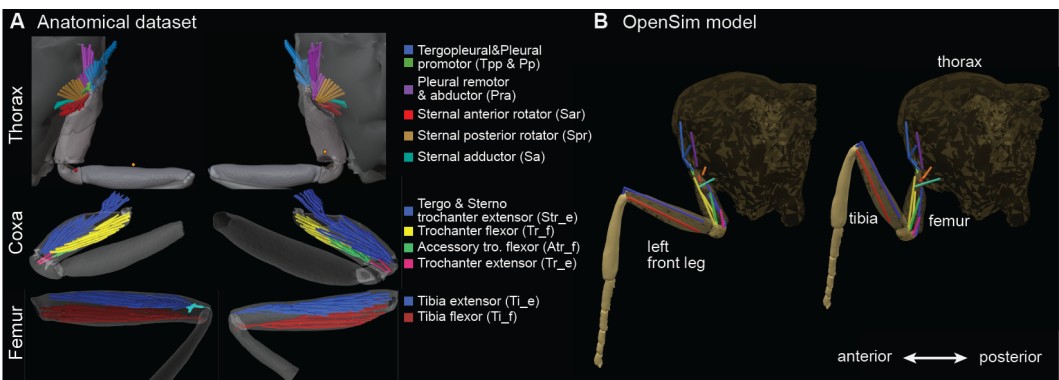

Figure 2: **Front leg muscle reconstructions. (A)** Muscles of the thorax, coxa, and femur were segmented from high-resolution X-ray scans (Kuan et al., 2020; Dinges et al., 2021) and visualized within a 3D mesh of the foreleg in Blender. Colors denote anatomically grouped muscles, including biarticular and joint-specific actuators. **(B)** Corresponding muscle-tendon units implemented in OpenSim preserve anatomical attachment points and fiber routing. Color coding is the same as in panel A.

### 3.2 MUSCLE MODEL CONSTRUCTION AND OPTIMIZATION

We modeled 15 muscle-tendon units (MTUs) per foreleg—7 in the thorax, 6 in the coxa, and 2 in the femur—capturing 12 of the 19 muscle groups in our anatomical datasets (Figure 2B). Muscles housed in the tibia were excluded because the tibia segment was partially captured in the X-ray data. In the femur, we modeled the fast tibia flexor and extensor, which dominate force generation at the femur–tibia joint (Azevedo et al., 2020). Trochanter muscles were omitted because their function remains unclear (Soler et al., 2004).

Each muscle group was modeled using one or two MTUs, based on a Hill-type formulation (Hill, 1938). This formulation includes a contractile element, a passive parallel elastic component, and a series elastic component, assuming rigid tendons (Millard et al., 2013) (Figure 3A). To keep the model computationally manageable in OpenSim (Delp et al., 2007), we selected up to two representative MTUs per muscle group to capture the main muscle fiber functions (see Section A.5 for more details). Muscle-tendon lengths and initial attachment points were initialized using anatomical data (Figure 3A, right). We estimated maximum isometric forces from the physiological cross-sectional areas (PCSAs) of the muscles, scaling them with a specific tension of $28\mathrm{mN/mm}^2$. This value falls between those reported for *Drosophila* jump muscles ($37\mathrm{mN/mm}^2$) (Eldred et al., 2010) and indirect flight muscles ($9\mathrm{mN/mm}^2$) (Swank, 2012). We estimated maximum contraction velocity using a recorded X-ray video of muscle contraction during leg movements.

Because experimental data are limited, initial parameters may not faithfully reflect biological reality. Therefore, we refined parameters through optimization in OpenSim. For each candidate parameter set proposed by the optimizer, static optimization (SO) inferred muscle activations from reference

joint angles, and forward dynamics (FD) simulated joint trajectories based on those activations. We used NSGA-II to identify parameter sets that produced kinematics best matching the experimental data (Figure 3B). To avoid overfitting, we optimized parameters to track kinematics of the front legs during two behaviors—antennal grooming and locomotion—retaining solutions that performed well for both behaviors (Figure 3C). Each joint (1 or 3 DoFs) was optimized independently. For a 3-DoF joint, optimization took approximately 8 h with 200 individuals over 40 generations. The full foreleg optimization (7 DoFs, 15 MTUs) would require $\sim$ 20 hours if each optimization were run sequentially using an Intel i9-14900 processor with 64 GB of RAM.

To assess the biological plausibility of our resulting muscle models, we analyzed muscle moment arms relative to joint motion. Our model correctly reproduced opposing signs for flexor and extensor moment arms, and predicted the dominant contributors to joint movements (thorax-coxa and coxa-trochanter yaw, pitch, and roll, and femur-tibia pitch) in agreement with their known functional roles (Figure 3D) (Azevedo et al., 2024; Soler et al., 2004).

### 3.3 REAL LIMB KINEMATICS DATA

2D and 3D limb kinematics were measured from tethered flies behaving spontaneously on an air-supported spherical treadmill. Foreleg and head movements were tracked in 2D using DeepLabCut (Mathis et al., 2018). Separate models were trained for each camera angle. Annotations were refined over multiple iterations to improve accuracy. Five synchronized cameras were calibrated using a ChArUco board, and 3D poses were reconstructed using Anipose (Karashchuk et al., 2021) with Viterbi filtering and spatiotemporal regularization. The resulting 3D pose data were then used to estimate leg and head joint angles via inverse kinematics using SeqIKPy (Özdil et al., 2026). Behavioral videos were recorded at 100 Hz and interpolated to 500 Hz to ensure simulation stability. We focused on two distinct behaviors, forward walking and antennal grooming, because they involve different patterns of joint coordination in the front legs and can thus test the capacity of our muscle model to reproduce a wider range of movements.

### 3.4 IMITATION LEARNING

To reproduce control strategies that animals use to drive efficient limb movements, we trained policies using imitation learning of reference motion trajectories. Imitation learning has been widely applied in both computer graphics and motor control, providing a powerful framework for learning complex behaviors in simulated agents (Vaxenburg et al., 2025; La Barbera et al., 2021; Chentanez et al., 2018; Peng et al., 2018; Merel et al., 2017; Peng et al., 2020; Hasenclever et al., 2020). Here, we trained neural network models to drive limb movements by activating muscles. Using MyoConverter (Caggiano et al., 2022), we converted our OpenSim model with optimized muscle parameters into a MuJoCo-compatible format. We then created the control task in the MuJoCo physics engine (Todorov et al., 2012), using `dm_control` (Tunyasuvunakool et al., 2020), to track the measured kinematics in a muscle-actuated manner. To improve speed and stability, we omitted body–environment contacts by suspending the fly in air.

We trained multilayer perceptron (MLP) policies with Proximal Policy Optimization (PPO) (Schulman et al., 2017) for $15 \times 10^6$ steps at a control frequency of 500 Hz, while running the physics engine at 10 kHz to ensure stability. We used PPO due to its well-known stability and robustness in high-dimensional continuous control tasks and based on its prior use in animal imitation learning tasks (Peng et al., 2018; 2020). The MLPs have 2 hidden layers of 512 units followed by a layer of 256 units each and use the ReLU activation function. Policies are trained with a learning rate of $10^{-5}$ and a discount factor of 0.99, chosen through hyperparameter search. Each training episode was initialized from a random frame in the motion capture sequence and terminated once the end of the clip was reached.

At each timestep, the agent received an observation including joint angles, 3D positions of selected body parts, muscle states, and the remaining time in the clip as described in (La Barbera et al., 2021). The observation space had 86 dimensions: 3D coordinates of four leg landmarks (12-D), joint angles and velocities for the 7 DoFs (14-D), and muscle activations, forces, lengths, and velocities for all 15 muscles (60-D). The policy produced continuous muscle excitation levels (i.e., motor neuron activities) within the range of $[0, 1]$ for each muscle (15-D).

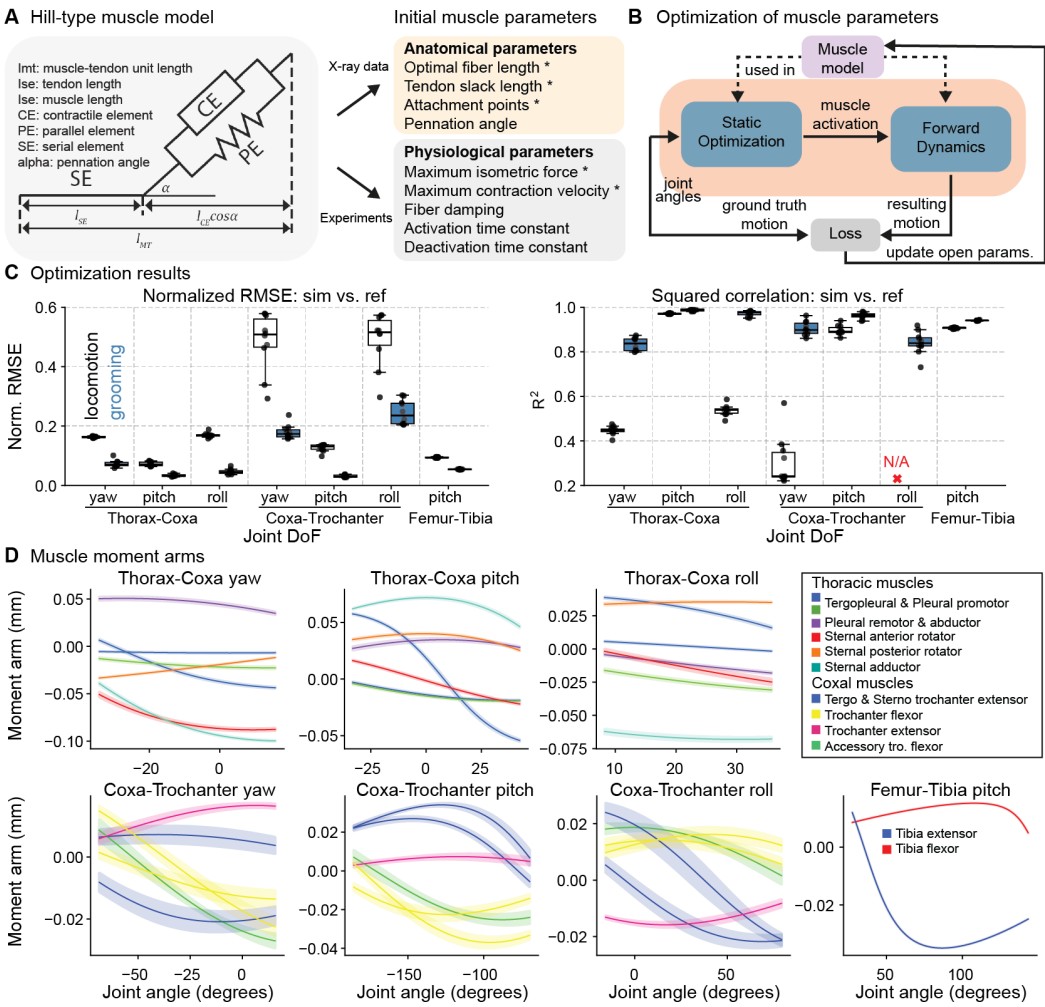

Figure 3: **Optimization and assessment of muscle model parameters.** (A) Schematic representation of a Hill-type muscle model. The contractile element (CE) produces active force, the parallel elastic element (PE) provides passive stiffness, and the series elastic element (SE) represents tendon elasticity. $l_m$ is the fiber length, $\alpha$ the pennation angle, and $l_t$ the tendon length. The total muscle-tendon length is $l_{mt} = l_m \cos(\alpha) + l_t$. Anatomical parameters (fiber length, attachment points, pennation angle) are initialized from X-ray data; physiological parameters (maximum isometric force, activation dynamics) from experiments and literature. Parameters indicated with asterisks are modified through optimization to match measured limb movements. Otherwise parameters were kept at fixed values. (B) Optimization pipeline for muscle parameters. Muscle parameters are jointly tuned to minimize the error between measured and simulated joint angles for two behaviors (antennal grooming and locomotion). (C) Performance of muscle parameter optimization. Left: root mean squared error (RMSE), normalized with respect to joint movement range. Right: square of the Pearson correlation between simulated and reference joint angles across 7 DoFs (3 joints). Because the coxa-trochanter roll joint does not move during locomotion, it is not shown and has constant values. Data from the top 10 individuals are shown and indicated (dots). (D) Moment arms of thoracic, coxal, and femoral muscles with respect to a single joint DoF. Other joints are fixed at mid-range positions. Plots show means and standard deviations across 10 individuals.

The reward function was designed to encourage accurate tracking of reference trajectories in Cartesian space, joint angle space, and joint velocity space. At each timestep, the reward was defined as:

$$r_t = \frac{1}{3}\left[ \exp(-w^p d_t^{xpos}) + \exp(-w^p d_t^{qpos}) + \exp(-w^v d_t^{qvel}) \right], \qquad (1)$$

where $d_t^{\text{xpos}}$, $d_t^{\text{qpos}}$, and $d_t^{\text{qvel}}$ denote the mean Euclidean distances between the simulated body and the reference trajectory in key point position, joint position, and joint velocity, respectively. The reward was clipped to the range $[0, 1]$. The weights were set to $w^{\text{p}} = 5$, and $w^{\text{v}} = 3$. To increase robustness, joint angles were initialized with white noise of variance 0.02. For a 7-DoF leg model with 15 MTUs, imitation learning took approximately 96 hours to complete on an Intel Core i7-12700 processor with 128 GB of RAM.

## 4 RESULTS

### 4.1 MUSCLE SYNERGIES DURING WALKING AND GROOMING

To investigate how individual muscles contribute to movement, we analyzed activation patterns during forward walking and grooming. Each walking trial contained three stance-swing cycles, and each grooming trial contained three bouts of leg sweeps (Figure 4A). Using OpenSim's static optimization, we estimated joint torques, muscle forces, and activations from our musculoskeletal model for both behaviors.

Among thoracic muscles, the pleural remotor abductor (Pra) was active at stance onset, consistent with its known role in initiating stance motion (Figure 4B, top) (Azevedo et al., 2024). The tergopleural promotor and pleural promotor were elevated during stance-to-swing transitions, driving the coxa forward (Figure 4B, top). During grooming, by contrast, the trochanter flexor and extensor showed rhythmic, overlapping activity, whereas they were out of phase during locomotion (Figure 4B, bottom).

How are muscles acting in fixed groups (i.e., synergies) to produce these movements? To address this question, we applied Non-negative Matrix Factorization (NMF), a widely used method for uncovering low-dimensional structure in muscle activity (Rabbi et al., 2020), to our muscle activity data. Remarkably, just three muscle primitives explained over 90% of the variance (Figure 4C); therefore, we proceeded with these first three primitives (see Section A.6 for more details). Each primitive exhibited distinct, non-overlapping temporal dynamics (Figure 4D). The first primitive alone captured more than 80% and aligned with stance onset during locomotion and half of the grooming cycle.

We further examined synergy weights to reveal both invariance and flexibility. Sar and Sa contributed consistently across all synergies and behaviors (Figure 4E), suggesting that they were task-invariant muscles. By contrast, coxal muscles showed task-specific specializations: flexors contributed primarily to synergy 2, whereas extensors contributed to synergy 3 for grooming. This specialization was absent in locomotor primitives (Figure 4E), suggesting that these muscles might have broader roles during locomotion.

Together, these findings predict that muscle coordination is highly behavior-dependent and that the fly might flexibly repurpose the same musculature by engaging distinct, task-specific synergies.

### 4.2 PASSIVE JOINT PROPERTIES FACILITATE MUSCLE-DRIVEN CONTROL

Movement is shaped not only by active muscle forces but also by passive biomechanical forces in the legs that can be either resistive or assistive (Ache and Matheson, 2013). These properties are thought to offload some control effort from the nervous system, suggesting that they may also facilitate learning in musculoskeletal models.

To test this hypothesis, we systematically varied the passive properties of our MuJoCo model's joints. We considered three parameters: stiffness, which produces spring-like forces that return joints to their reference angles; damping, which resists velocity; and armature, which adds joint inertia (Figure 5A). Armature was included in all conditions to stabilize simulations, while stiffness and damping were selectively added or removed, yielding four different biomechanical models (Figure 5B). Each model was then trained with PPO to imitate locomotor kinematics for 15 million steps. We quantified learning performance by measuring the average reward early (2.5-3M steps) and late (14.5-15M steps) in training (Figure 5C).

We found that the combination of stiffness and damping yields the fastest learning and the highest performance, outperforming models with either property alone (Figure 5D, left). This advantage

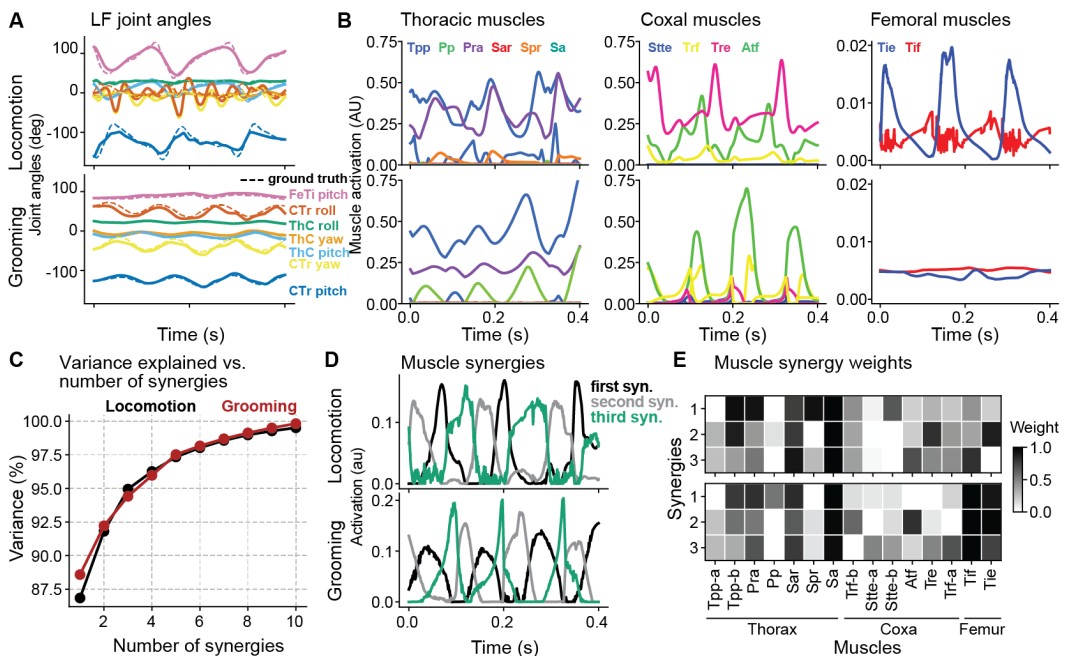

Figure 4: **Predicting muscle synergies from simulated muscle activations in OpenSim.** **(A)** Simulated joint angle trajectories of the left foreleg (LF) during locomotion (top) and grooming (bottom), obtained from the Static Optimization-Forward Dynamics pipeline. The dashed lines indicate the ground truth joint angles. **(B)** Simulated muscle activation dynamics for thoracic (left), coxal (middle), and femoral (right) muscles during locomotion (top) and grooming (bottom). **(C)** Variance explained by increasing numbers of synergies for locomotion (black) and grooming (red). Three primitives capture over 90% of the variance. **(D)** Time courses for the first three extracted synergies (black, gray, and green). **(E)** Synergy weight matrix showing each muscle's contribution; darker values indicate stronger loading.

persisted to the end of training (Figure 5D, right), although final kinematics were qualitatively indistinguishable across models (Figure 5E). We next asked how passive properties shape muscle control. Averaging muscle activations across time and seeds revealed similar mean levels across conditions, but with different temporal dynamics (Figure 5F, Figure S7).

Overall, these results support the idea that passive joint mechanics can substantially improve the learnability and robustness of muscle-driven control. We speculate that this is because they stabilize motion and reduce the need for constant correction, allowing the policy to focus on producing effective coordinated muscle patterns rather than correcting instability. In biology, a similar division of labor between passive mechanics and active control might support movement efficiency and robustness (Ache and Matheson, 2013).

## 5 DISCUSSION

Here, we present an end-to-end pipeline for constructing morphologically realistic Hill-type muscle models based on annotated anatomical data. Our framework extracts each muscle's anatomical features, estimates physiological parameters, and optimizes unknown parameters using multiobjective optimization. To our knowledge, a similarly comprehensive and automated approach linking anatomical inputs to functional, muscle-driven simulations is currently lacking in the field, particularly for widely used model organisms like *Drosophila melanogaster*.

We modeled 15 muscle-tendon units (MTUs) per front leg in NeuroMechFly, a biomechanical model of adult *Drosophila* (Lobato-Rios et al., 2022; Wang-Chen et al., 2024). After optimizing muscle parameters, we evaluated how each MTU contributes to different joint DoFs during locomotion and grooming. Our results predict that grooming and locomotion employ distinct muscle synergies.

In addition, we show that incorporating passive joint properties —such as damping and stiffness— provides useful priors for musculoskeletal control, improving learning efficiency. Together, these results demonstrate that combining detailed musculature with biologically inspired joint mechanics enables fast, robust, and scalable simulations of muscle-driven behavior and facilitates the efficient training of neural controllers.

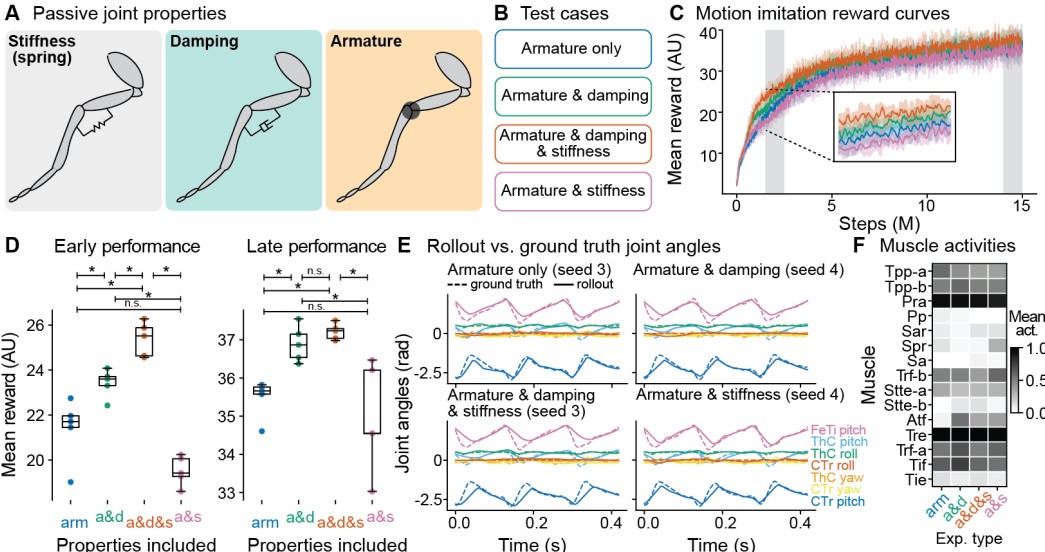

Figure 5: **The impact of passive joint properties on imitation learning of limb kinematics. (A)** Schematic of passive joint properties modified in MuJoCo: stiffness (spring), damping, and armature (inertia). **(B)** Experimental conditions combining these properties: (i) armature only, (ii) armature and damping, (iii) armature and damping and stiffness, and (iv) armature and stiffness. **(C)** Motion imitation rewards at the evaluation stage over the training period, averaged across 5 random seeds (lines) with 5th-95th percentile ranges (shaded regions). Gray boxes indicate evaluation windows for early and late performance in panel D. **(D)** Mean rewards at early (left) and late (right) training stages. Each point is one seed. Statistical comparisons used two-sided Mann-Whitney U tests with Holm-Bonferroni correction (* $P < 0.05$, ns: not significant). **(B-D)** Color code is the same across panels. **(E)** Joint angle trajectories from ground-truth kinematics (dashed) and policy rollouts (solid) for each condition for seeds with the highest final reward. **(F)** Time-averaged muscle activities across conditions; darker shading indicates higher activation.

Our work provides a critical interface between the outputs of neural network controllers and physical movements. In our framework, the possible actions taken by the policy sit in a space that, through the musculoskeletal model, can only generate physically plausible movements. This eliminates many unrealistic or even catastrophic actions that would otherwise burden the learning process. In other words, whereas a model that naively controls target joint states would have to first learn (explicitly or implicitly) a world model of physics before learning a policy that operates within physical constraints, our model only needs to learn the latter. This approach enables efficient training at scale, offering a promising path to reduce the sim-to-real gap (Wechsler et al., 2024) without requiring costly task-specific real-world data. Furthermore, integrating state-dependent exploration strategies like DEP-RL (Schumacher et al., 2022) could further enhance sample efficiency in these high-dimensional muscle action spaces. Our work focuses on *Drosophila*, an animal with the most complex leg kinematics whose entire nervous system has been mapped. This opens the door to testing neural architectures which are optimized through evolution in embodied settings. We believe our work can impact neuroscience, robotics, and machine learning, areas that share a common focus of understanding and replicating motor control in physically grounded systems.

## 6 LIMITATIONS AND FUTURE WORK

Our model has several limitations that can be addressed in future work. One major challenge is the scarcity of experimental data and high uncertainty in our chosen physiological parameters. For example, key properties such as the maximum isometric forces and contraction velocities were not directly measured but instead were estimated and optimized using a combination of anatomical and physiological data. Our model will benefit from the acquisition of further experimental data that will allow users to narrow down the space of possible parameter values.

Another limitation is that body–body and body–environment contact forces were omitted during training. To assess robustness, we rolled out the trained policy with ground contacts enabled using a hybrid controller in which the left foreleg was muscle-driven while the remaining legs were position-controlled using recorded locomotion kinematics. Remarkably, without any additional tuning, the muscle-driven leg generated sufficient forces to maintain a stable posture and propel the body forward (Figure S8).

Despite these limitations, our work represents the first biologically-grounded muscle modeling framework for studying motor control in adult *Drosophila melanogaster*. It enables researchers to test hypotheses of muscle function and provides a foundation for uncovering unmodeled passive dynamics and emergent biomechanical properties. Integrating this framework with emerging experimental datasets, such as *in vivo* muscle imaging (Melis et al., 2024), will help refine physiological and anatomical constraints, narrow down the parameter space, and thereby improving the predictive power of musculoskeletal models.

## CODE AND DATA AVAILABILITY

All simulation videos and links to the full source code required to reproduce the results are publicly available at the project website: `https://gizemozd.github.io/fly_mimic`.

## ACKNOWLEDGEMENTS

We thank Alexander Blanke for providing the *Drosophila* X-ray dataset. P.R. acknowledges support from a Swiss National Science Foundation (SNSF) Project Grant (175667) and a SNSF Eccellenza Grant (181239). A.I. and C.N. were funded by the ERC Synergy Salamandra grant (No 951477). S.W.-C. acknowledges support from a Boehringer Ingelheim Fonds PhD fellowship. J.S.P. was supported by postdoctoral fellowships from EMBO (award ALTF 762-2022), HFSP (award LT0029/2023-L), and the Helen Hay Whitney Foundation. P.G.Ö. acknowledges support from a Swiss Government Excellence Scholarship for Doctoral Studies and a Google PhD Fellowship.

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

## A    SUPPLEMENTARY MATERIAL

### A.1    MUSCLE RECONSTRUCTION AND MODELING OF THE MIDDLE AND HIND LEGS

We concentrated our efforts on developing a fully functional front-leg muscle model for two main reasons: (i) we have access to a rich kinematic dataset spanning diverse behaviors such as grooming and locomotion, and (ii) multiple complementary datasets exist that validate joint rotation centers and degrees of freedom (DoFs).

Nevertheless, we also annotated muscle fibers in the midlegs (Figure S1A) and hindlegs (Figure S2A) using our custom X-ray dataset. In total, we modeled seven muscle–tendon units (MTUs) per midleg (Figure S1B) and eight MTUs per hindleg (Figure S2B). Because only a single dataset was available for these legs, our ability to examine muscle attachment points across joint configurations or to cross-validate the reconstructions was limited. For this reason, we restricted our work to anatomical reconstruction rather than optimizing the models to reproduce motor behaviors.

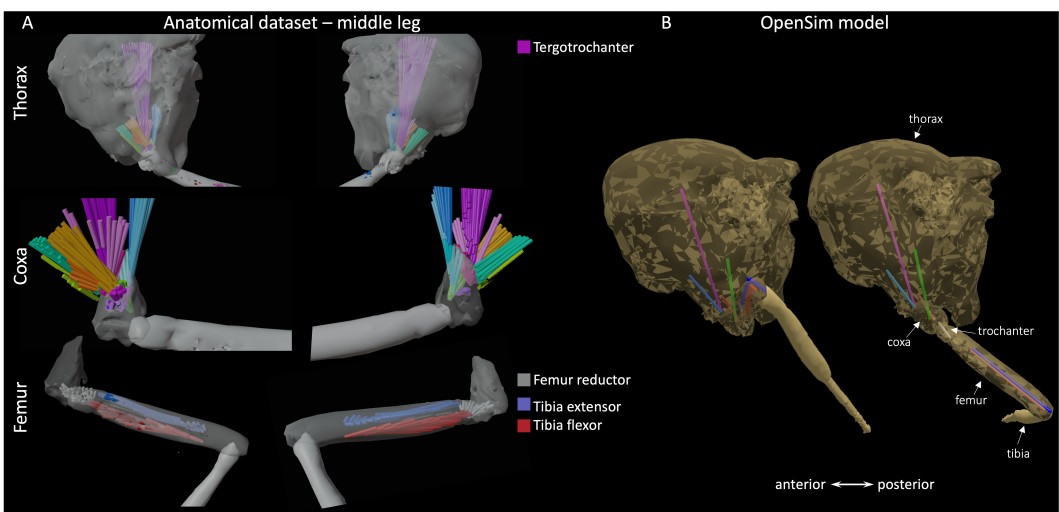

Figure S1: **Middle leg muscle reconstructions. (A)** Muscles of the thorax, coxa, and femur were segmented from a high-resolution X-ray scan and visualized within a 3D mesh of the leg in Blender. Distinct colors denote anatomically grouped muscles. **(B)** Corresponding muscle-tendon units implemented in OpenSim preserve anatomical attachment points and fiber routing. Color coding is the same as in panel A and reflects functionally grouped muscles.

Future datasets capturing the mid- and hindlegs at high resolution will enable more detailed biomechanical modeling and muscle reconstructions. Once such data become available, our framework can be readily extended to achieve muscle-driven control of all six legs.

### A.2    REFINING THE BIOMECHANICAL MODEL

A fundamental challenge in muscle modeling is accurately grounding muscle attachment points and joint movements in anatomical data. To better align our model with biological reality, we replaced NeuroMechFly's foreleg meshes with those derived from X-ray scans (Figure S3A), where muscles were reconstructed. This adjustment was motivated by two key factors. First, using the same leg mesh as the dataset ensured a one-to-one mapping between the muscles in the dataset and our model. Second, we identified a major discrepancy in the fully flexed resting posture of the original model's leg compared to real fly anatomy. Specifically, in the experimental data, the trochanter segment of the front leg forms a bridge between the coxa and femur, positioning them side by side rather than stacked vertically when fully flexed (Figure S3B). As a result, when fully extended, the foreleg adopts a parenthesis-like shape rather than forming a straight line (Figure S3A).

With this improved biomechanical model, we next investigated the joint rotational axes and centers that could more accurately replicate real leg movements. To determine these parameters, we exam-

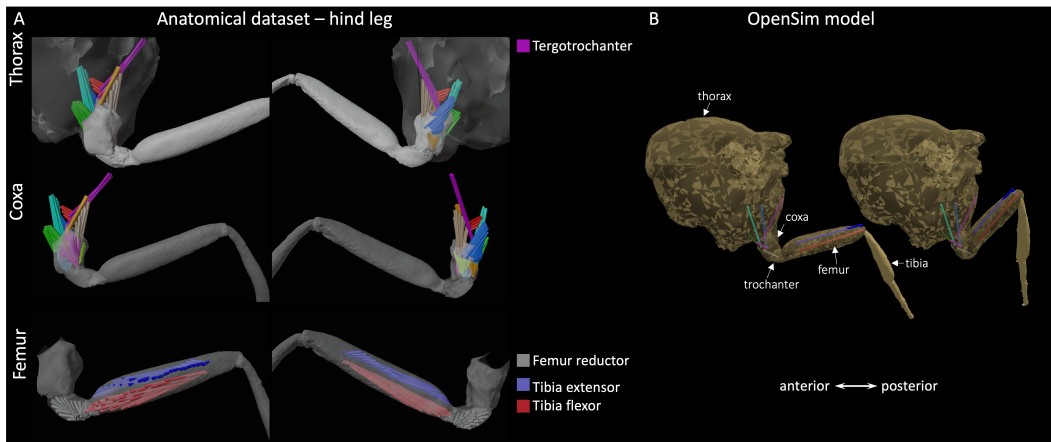

Figure S2: **Hind leg muscle reconstructions.** **(A)** Muscles of the thorax, coxa, and femur were segmented from a high-resolution X-ray scan and visualized within a 3D mesh of the leg in Blender. Distinct colors denote anatomically grouped muscles. **(B)** Corresponding muscle-tendon units implemented in OpenSim preserve anatomical attachment points and fiber routing. Color coding is the same as in panel A and reflects functionally grouped muscles.

ined muscle attachment points and joint condyles (Haustein et al., 2024). Our analysis suggested that the coxa-trochanter joint may, in fact, possess three DoFs, in contrast to the originally assumed two DoFs (Lobato-Rios et al., 2022).

To test this hypothesis, we used SeqIKPy (Özdil et al., 2026) to design a kinematic chain with three DoFs at the coxa-trochanter joint. The revised joint configuration was equally effective at tracking the leg trajectory compared to the original two-DoF model (Figure S3C). However, notably, the three-DoF configuration reduced the reliance on thorax-coxa joint rotations during antennal grooming, decreasing the range of motion from $[-95°, 50°]$ to a more constrained interval of $[-30°, -7°]$ (Figure S3D), preventing unnatural rotations of the thorax-coxa joint.

### A.3 HILL-TYPE MUSCLE MODEL

We chose to use a Hill-type muscle model because it offers a good balance between biological realism and computational efficiency. Hill-type muscle models are widely adopted in the biomechanics and neuromechanics communities due to their versatility and biological plausibility (Caillet et al., 2022). By contrast, Ekeberg-type muscle models are computationally very efficient yet have limited biological correspondence (Ekeberg et al., 2004). Compared with simpler torque-driven or mass-spring systems, Hill-type models explicitly represent the key components of a muscle-tendon unit (MTU)—namely, contractile, series elastic, and parallel elastic elements. This structure allows us to capture essential nonlinear properties of biological muscles, including force-velocity and force-length relationships, which are critical for simulating realistic motor behavior.

We adopted a Hill-type model based on (Millard et al., 2013) and (Geyer et al., 2003), consisting of a Contractile Element (CE) for active force generation, Parallel Elasticity (PE) for passive stiffness, and Series Elasticity (SE) representing the tendons. The Buffer Elasticity (BE) was omitted due to its negligible contribution to muscle dynamics (Geyer and Herr, 2010).

The total muscle force is given by:

$$F_{MT} = (F_{CE} + F_{PE} + F_{\text{damper}}) \cos \alpha = F_{SE} \qquad (S1)$$

where $F_{CE}$, $F_{PE}$, and $F_{\text{damper}}$ are the active, passive, and damping forces, respectively. The active force is computed as:

$$F_{CE} = a(t) F_{\max} f_l(l_{CE}) f_v(v_{CE}) \qquad (S2)$$

where $a(t)$ is the activation state, $F_{\max}$ the maximum isometric force, and $f_l(l_{CE})$ and $f_v(v_{CE})$ the force-length and force-velocity relationships, respectively. The passive force $F_{PE}$ is only active during muscle elongation, while the damping force depends on contractile element velocity.

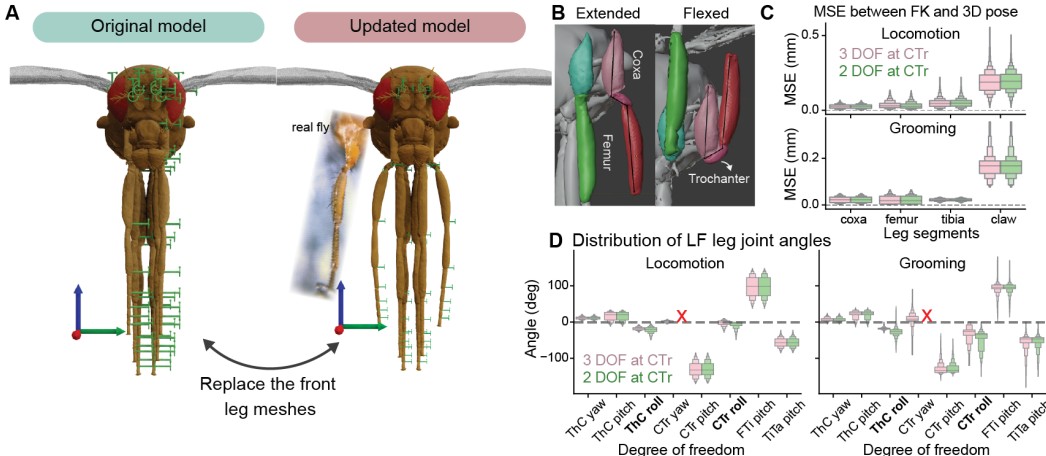

Figure S3: **Enhancements in the biomechanical model of NeuroMechFly. (A)** The forelegs of NeuroMechFly were updated by replacing the original meshes with an anatomically detailed muscle-based dataset. Unlike the previous model, where the fully extended leg was nearly straight, the updated model exhibits a natural parenthesis-like curvature, consistent with the biological data (middle). **(B)** The trochanter's role in leg posture. The trochanter bridges the coxa and femur, positioning them side by side rather than in a linear alignment. In the previous model (green), this structural feature was simplified, whereas the updated model (red) better captures the fly's natural joint configuration. **(C)** Mean squared error (MSE) between measured 3D poses and forward kinematics (FK) reconstructed from inverse kinematics, comparing 2-DoF and 3-DoF coxa-trochanter joint configurations during locomotion and grooming. Both kinematic configurations perform equally well. **(D)** Distribution of left foreleg joint angles during locomotion and grooming for different degrees of freedom, showing the effect of anatomical refinements and DoF addition on the joint angle range. ThC roll and CTr roll (in bold) show a visible decrement in range in the 3-DoF model.

For compatibility across simulation engines, we set the pennation angle to zero and scaled $F_{\max}$ accordingly. Assuming a rigid tendon, the model state variables reduce to the muscle fiber length $l_{CE}$, ensuring $C^2$-continuity. The complete model is defined by physiological parameters (e.g., $F_{\max}, v_{\max}, \alpha, \tau_{act}, \tau_{deact}$) and anatomical parameters (e.g., $l_{opt}, l_{slack}$, muscle attachment points, and muscle paths), detailed in Table 1.

We modeled each functional muscle group as one MTU. Unlike human muscles, many fly muscles attach directly to the inner cuticle, often without tendons, and fibers within a group can vary considerably. To model each group, we selected representative fibers using the following criteria: (i) If the muscle attaches via a tendon, we used the tendon's location. If attachment is via multiple fibers, we selected the fiber closest to the group's center of mass—typically also the median-length fiber. (ii) For groups with widely spread attachment points, we subdivided them into smaller, more homogeneous clusters and selected one representative fiber per cluster.

## A.4 MUSCLE MODELING IN OPENSIM

Muscles were defined as *forceset* objects in OpenSim, serving as force generators. The Millard 2013 muscle model (Millard et al., 2013) was used, with parameters categorized into three types (Table 2): (i) open parameters requiring optimization, (ii) fixed parameters set by model choice, and (iii) default parameters left unchanged.

Most anatomical and physiological parameters were open parameters, extracted or estimated through data-driven methods described in the methods section of this paper. Assuming rigid tendons, tendon compliance was ignored, and pennation angles were set to zero, meaning MTU length changes were solely due to muscle fiber length changes.

Control and activation parameters were adjusted for greater flexibility, with a minimum activation of 0.01%, while max control and detailed F-L and F-V curve parameters were left at their default

Table 1: A collection of model parameters used in the muscle model. Letters A and P stand for anatomical and physiological parameters, respectively.

| Parameter | Type | Description |
|---|---|---|
| Optimal fiber length | A | The fiber length at which actin–myosin interactions are maximal, i.e., when maximum force can be produced. |
| Tendon slack length | A | The length beyond which a muscle's tendons begin resisting stretch and producing force. |
| Pennation angles | A | The angle between a fascicle's orientation and the tendon axis. |
| Muscle cross-section area | A | The cross-section area of muscle fibers. |
| Muscle attachment points | A | The points to which a muscle-tendon unit connects at its fiber start (origin) and tendon end (insertion). Optionally an additional attachment point can be defined for more complex muscle paths |
| Muscle paths | A | The path that a muscle-tendon unit travels along from its origin to its insertion. |
| Maximum isometric force | P | The maximum isometric force generated by the Contractile Element (CE) at its optimal length $l_{opt}$. |
| Maximum contraction velocity | P | The maximum velocity at which the Contractile Element (CE) can contract. |
| Activation time constants | P | A time constant determining the transfer speed from neural signal to muscle activation. |
| Deactivation time constants | P | A time constant determining the transfer speed from neural signal to muscle deactivation. |
| Muscle damping factors | P | The coefficient for the muscle damping force as defined in Millard's muscle equation. |

values. If not explicitly defined, these curves followed empirical formulations (Millard et al., 2013). Due to the lack of measured *Drosophila* muscle curves, default curves in OpenSim were assumed to approximate real physiological behavior.

## A.5   MUSCLE PARAMETER OPTIMIZATION USING NSGA-II IN OPENSIM

To optimize muscle parameters, we used NSGA-II, an elitist genetic algorithm well-suited for multi-objective optimization problems (Deb et al., 2002), implemented in Python using the `geatpy` package.

Each muscle–tendon unit (MTU) was optimized in a 6-dimensional parameter space (or 9-dimensional when an additional via point was included in the muscle path), with separate objective metrics defined for each active degree of freedom (DoF). These objectives aimed to minimize both the mean squared error and the reverse correlation between the ground-truth and simulated motion. To reduce overfitting of muscle parameters, we ran the SO-FD pipeline for both locomotion and grooming behaviors (Figure S4). The total objective score was computed by summing the two scores for each DoF. NSGA-II hyperparameters were empirically tuned (Table 3), and parameter search ranges were iteratively refined based on optimization performance. The distribution of the resulting muscle parameter values is shown in Figure S5.

Specifically, the following parameters were optimized:

- Maximum isometric force was computed as the product of a fixed base tension (from prior experimental work), a scaling factor (optimized between 0.3–3), and the physiological cross-sectional area (PCSA) calculated from CT scans.

- Maximum contraction velocity was defined as a base value (estimated from femur–tibia motion videos under X-ray) scaled by a factor between 0.4 and 2.4.

Table 2: Millard 2013 muscle model parameters categorized by type and optimization.

| Parameter | Value Type | Parameter Type |
|---|---|---|
| Optimal fiber length | Anatomical | Open |
| Tendon slack length | Anatomical | Open |
| Geometry path | Anatomical | Open |
| Max isometric force | Physiological | Open |
| Max contraction velocity | Physiological | Open |
| Activation time constant | Physiological | Fixed |
| Deactivation time constant | Physiological | Fixed |
| Ignore tendon compliance | Physiological | Fixed |
| Pennation angle at optimal | Anatomical | Fixed |
| Maximum pennation angle | Anatomical | Fixed |
| Min activation | Physiological | Fixed |
| Min control | Physiological | Default |
| Max control | Physiological | Default |
| Fiber damping | Physiological | Default |
| Active force–length curve | Physiological | Default |
| Force–velocity curve | Physiological | Default |
| Fiber force–length curve | Physiological | Default |
| Tendon force–length curve | Physiological | Default |

- Optimal fiber length and tendon slack length were based on ratios observed in the CT data, scaled by a factor between 0.8 and 1.2, and capped at a maximum of 95% of the total length.

- Muscle paths were allowed to vary within a cube of 5-10$\mu$m around the annotated insertion points.

Table 3: NSGA-II parameters used in optimization.

| Muscles | Generation | Population number | Mutation | Cross-over |
|---|---|---|---|---|
| Thorax | 200 | 120 | 0.7 | 0.5 |
| Coxa | 200 | 40 | 0.7 | 0.5 |
| Femur | 200 | 300 | 0.5 | 0.3 |

We applied a curriculum learning strategy to each joint using front leg kinematics during grooming and locomotion. The warm-up stage (5–10 generations) used conservative hyperparameters to encourage rapid convergence, followed by a fine-tuning stage (5–10 generations) that employed more exploratory search settings. Final optimized muscle parameters were validated in the SO-FD pipeline, and the best-performing set was mirrored to the right leg, completing the musculoskeletal model of the front legs.

A.6    RECONSTRUCTION OF MUSCLE ACTIVITY PATTERNS USING REDUCED SYNERGY SETS

To what extent can the recovered muscle synergies serve as effective control signals? To address this question, we reconstructed muscle activation patterns using varying numbers of synergies and used these reconstructions to drive all foreleg muscles in our model. Although one or two synergies captured more than 85% of the variance in the original muscle activity data (Figure 4C), the resulting joint kinematics deviated substantially from the ground truth (Figure A.6A,B). In contrast, reconstructions based on the top three synergies produced joint trajectories that qualitatively matched the measured kinematics (Figure A.6C). Interestingly, grooming kinematics were recovered more accu-

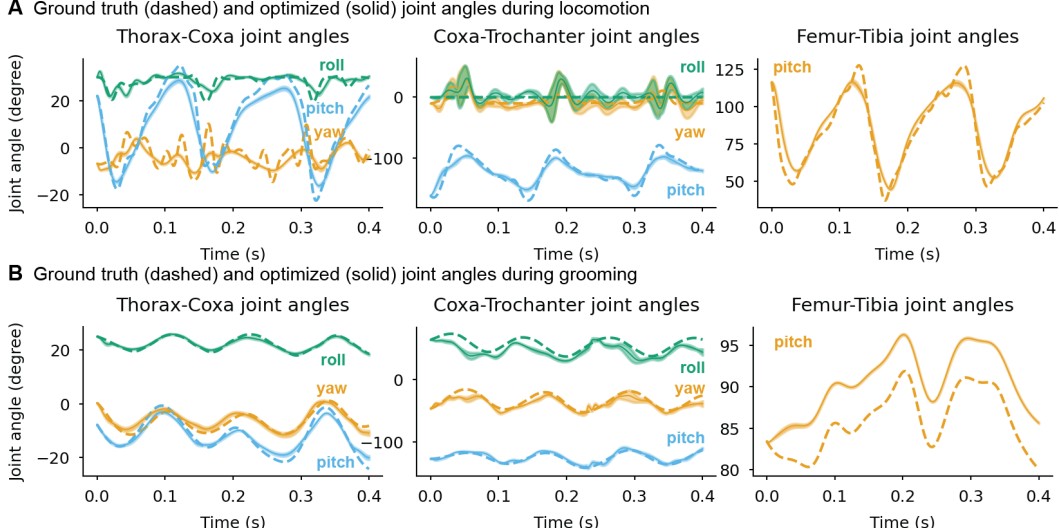

Figure S4: **Joint kinematics across all degrees of freedom in reference and simulated data.** Joint angle trajectories from **(A)** locomotion and **(B)** grooming behaviors are used as reference inputs for optimizing unknown muscle parameters. The MSE and correlation between the reference and resulting trajectories are used as objectives. The thick solid line represents the reference trajectory, the thin solid line shows the trajectory of the best-performing individual from the optimization, and the shaded region indicates the standard deviation across the top 10 individuals.

rately with fewer synergies compared to locomotion (Figure A.6D), suggesting that grooming may rely on a more synergistic and potentially lower-dimensional control algorithm.

### A.7 IMITATION LEARNING USING PPO IN MUJOCO

We trained multilayer perceptron (MLP) policies with Proximal Policy Optimization (PPO) (Schulman et al., 2017) for $15 \times 10^6$ steps at a control frequency of 500 Hz, while running the physics engine at 10,000 Hz to ensure stability. The hyperparameters used for training are listed in Table 4.

Initially, our motion capture dataset included only joint angles and velocities. To enrich this dataset, we replayed the joint kinematics using a point-torque model and recorded all observable physical quantities, including body positions and velocities in Cartesian space. This expanded dataset was then used as the reference trajectory in our imitation learning experiments.

At each timestep, the agent received an observation vector comprising joint angles, 3D positions of selected body parts (i.e., tibia-tarsus and claw), muscle states (i.e., length, velocity, activation, and force), and the remaining time in the clip, as described in (La Barbera et al., 2021). The policy output consisted of continuous muscle input values in the range $[0, 1]$ for each muscle.

The reward function was designed to encourage accurate tracking of the reference motion, both in joint space and Cartesian space. We initially tested the reward function and training setup in a point-torque model. Upon achieving successful performance in this simpler setting, we transitioned to the more complex musculoskeletal model. All parameter values were selected through a hyperparameter search.

### A.8 GROUND LOCOMOTION USING HYBRID MUSCLE- AND PD-CONTROL

We implemented a hybrid locomotion setup in which the left foreleg (LF) was driven by muscle activations and all remaining legs were controlled via PD-based position tracking. Fifteen LF muscles received time-varying activations derived from the trained policy (time step upsampled from 0.002s to 0.0001s). For the other five legs, desired joint angles were obtained from experimental kinemat-

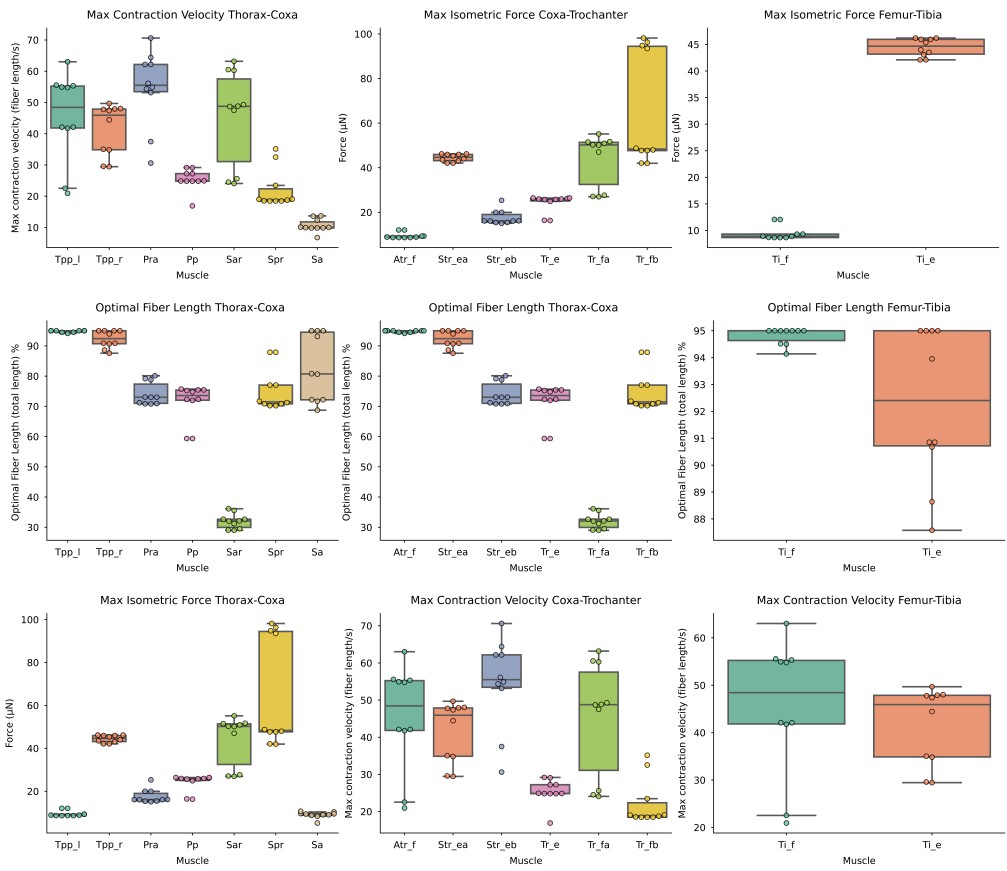

Figure S5: **Distribution of optimized muscle parameters.** Final values for maximum contraction velocity (top), optimal fiber length (middle), and maximum isometric force (bottom) after optimization. Each dot represents one of the top 10 individuals selected from a total population of 200. Muscles are grouped by segment (thorax, coxa, femur) from left to right.

Table 4: Key hyperparameters used for imitation learning with PPO. The same values were applied to both the actor and critic networks.

| Hyperparameter | Value |
| --- | --- |
| Network Architecture | [512, 512, 256] |
| Activation Function | ReLU |
| Optimizer | Adam |
| Learning Rate | $1 \times 10^{-5}$ |
| Batch Size | 64 |
| Rollout Length ($n_{\text{steps}}$) | 2048 |
| Epochs per Update | 10 |

ics synchronized with the trained and tracked with a PD controller (nominal gains: $K_p = 800$ and $K_d = 5$, with reduced gains).

Ground-reaction forces were extracted at each time step by summing the magnitudes of contact forces in the tarsal collision geometries for each leg. The resulting per-leg force traces were smoothed via a moving average filter with window size of 0.01 s and a third-order Savitzky-Golay filter with window size of 0.03 s (Figure S8).

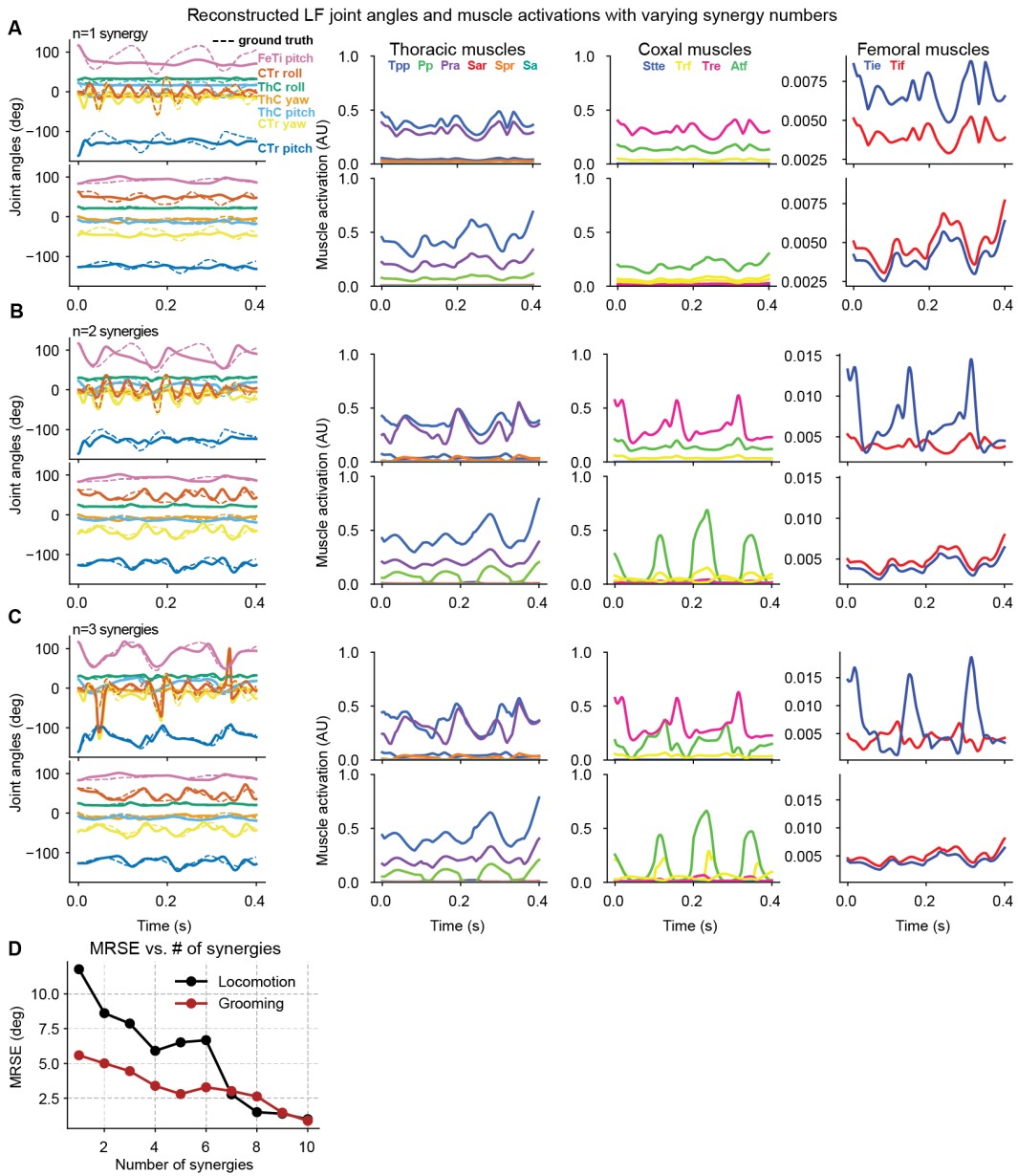

Figure S6: **Reconstructed left foreleg (LF) joint kinematics and muscle activations using varying numbers of muscle synergies.** **(A–C)** Reconstructed left foreleg joint angle trajectories (first column) and muscle activation dynamics for thoracic (second column), coxal (third column), and femoral (fourth column) muscles using **(A)** n=1, **(B)** n=2, and **(C)** n=3 synergies. Reconstructed joint angles are shown alongside ground-truth kinematics (dashed lines) for comparison. **(D)** Mean root-square error between reconstructed and ground-truth joint angles as a function of the number of synergies for grooming (red) and locomotion (black). Increasing the number of synergies improves reconstruction accuracy and yields more realistic muscle activation profiles across all muscle groups.

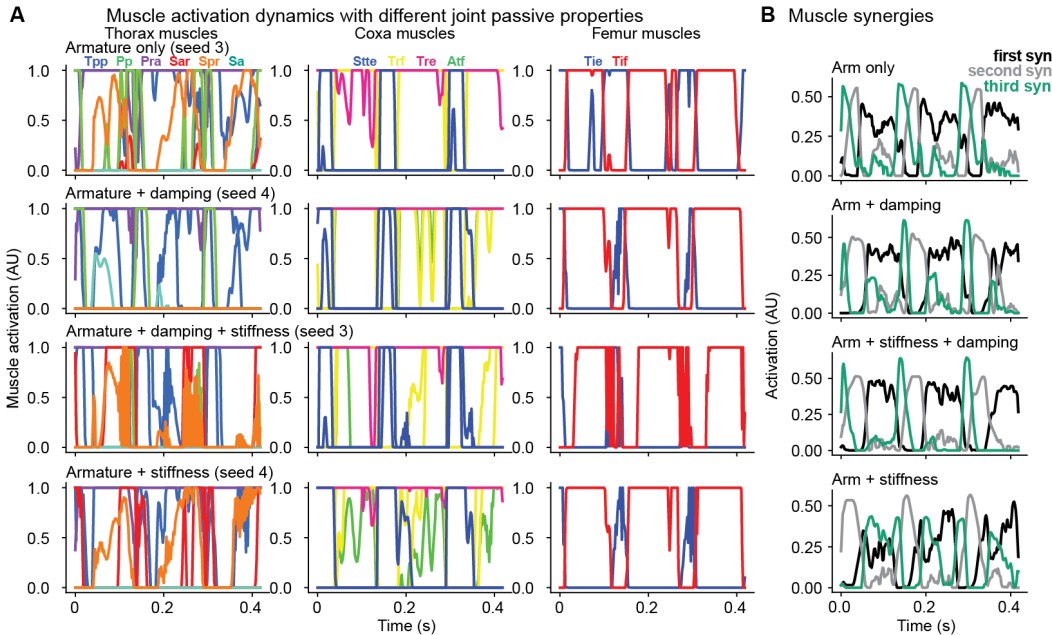

Figure S7: **Muscle activation dynamics across passive joint conditions.** **(A)** Simulated activation patterns of thorax (left), coxa (middle), and femur (right) muscles in PPO agents trained with varying passive joint properties: armature only (top), armature + damping (second row), armature + damping + stiffness (third row), and armature + stiffness (bottom). Each panel shows muscle activations from the best-performing seed for that condition. **(B)** Corresponding time courses of the first three extracted muscle synergies (black, gray, green), computed from the activation patterns shown in panel A.

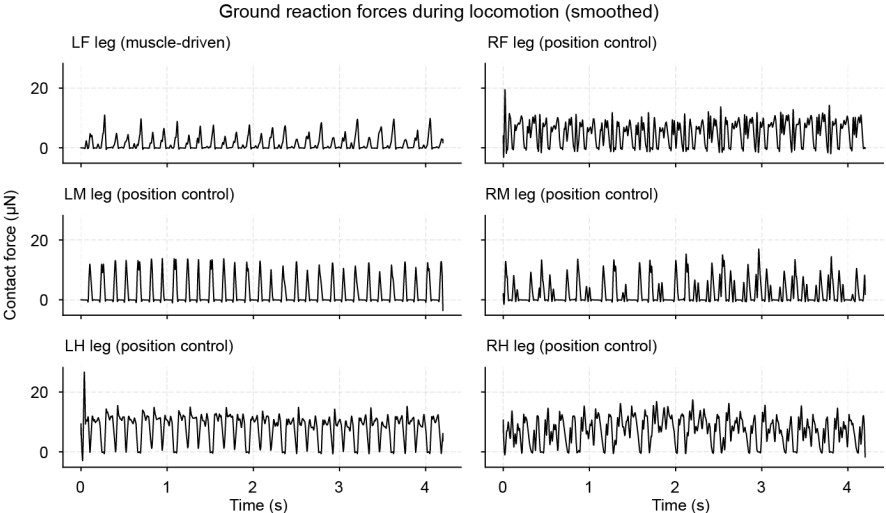

Figure S8: **Ground reaction forces during contact-enabled locomotion.** Smoothed ground reaction forces measured at each leg during locomotion with body-environment contacts enabled. The left foreleg (LF) is driven by the muscle activations produced by the trained policy (armature & stiffness & damping), while the remaining five legs (RF, LM, RM, LH, RH) are controlled via position controllers. Peaks in the force traces correspond to stance phases. Plotted values represent the norm of the ground reaction forces across the X, Y, and Z axes.

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
