# OpenReview forum: "Musculoskeletal simulation of limb movement biomechanics in Drosophila melanogaster"
_ICLR.cc/2026/Conference — ICLR 2026 Poster_

### Official Review · Reviewer_Lmjv · 2025-10-30

**Soundness:** 3
**Presentation:** 3
**Contribution:** 3
**Rating:** 6
**Confidence:** 4

**Summary:**

This paper constructs a foreleg musculoskeletal systems of fly based on anatomical data and optimization. The authors model 15 muscle-tendon units of each leg based on two public and one custom anatomical dataset, and optimize the muscle parameters to align with fly joint movement on OpenSim. They also convert the OpenSim model to the Mujoco platform, and augment the model stability by adding passive biomechanical forces. Model analysis demonstrate the qualitative accuracy of the fly musculoskeletal model in OpenSim, and the converted Mujoco model with passive forces is capable of imitating fly motion data.

**Strengths:**

1. The proposed model is the first musculoskeletal model of the fly, which requires substantial effort for the anatomical data. The model provides a platform for better biomechanics and neuroscience research.

2. The whole model building pipeline is detailed, which might help the musculoskeletal modeling of other animals.

3. The paper is clear and well written.

**Weaknesses:**

1. This model only has foreleg muscles and ignores all the contacts, which cannot achieve common behaviors such as locomotion and manipulation.

2. The whole model building, optimization and analysis is conducted in OpenSim, but the musculoskeletal control is conducted over Mujoco. The Mujoco model has passive biomechanical forces where the OpenSim model does not have. It is not clear whether the muscle accuracy and synergies result in Opensim can be transfered to a different platform.

3. In figure 3c, the RMSE of coxa-trochanter is large with low correlation. Video results on the tracking performances over both OpenSim and Mujoco might better help assessing the model building fidelity.

**Questions:**

1. In figure 3c and Figure 4d, what does (au) means in the y axis label?

2. In figure 4a, it would be better to show the reference joint trajectory compared against the joint movement generated by OpenSim static optimization.

3. How to determine the parameters of passive forces in Mujoco?

---

> ### Author Response · Authors · 2025-11-27
>
> We thank the Reviewer for their positive assessment of our work.
>
> **Response:**
>
> 1. In the revised manuscript, **we now enable full contact interactions between the fly and the environment for locomotion.** Because our middle and hindleg muscle reconstructions are not yet fully optimized, we controlled those legs (and the right foreleg) using position controllers, while actuating the left foreleg with the muscle activations produced by the trained policy from the armature & damping & stiffness model. Remarkably, without any additional training, the muscle-driven left foreleg generated sufficient forces to maintain stable stance and support ground contact.
>
> We have added a new supplementary video demonstrating ground locomotion with contact enabled, along with the corresponding ground-reaction forces for all six legs (five PD-controlled and one muscle-driven). We are currently working on optimizing the muscle model parameters for the middle and hind legs to achieve a fully muscle-driven multi-leg locomotion.
> We modeled the anatomical pathways of the middle and hind leg muscles (Figures S1 and S2). However,  we have not yet optimized their muscle parameters due to limited biomechanical data on those legs, which would require other X-ray datasets, including those legs. Having said that, our pipeline is fully compatible with the extended model. As more data becomes available, it can support full-body musculoskeletal simulations.
>
> 2. We agree that using different platforms for optimization (OpenSim) and RL training (MuJoCo) may introduce discrepancies. To minimize these effects, we took two steps. First, we used the OpenSim MillardEquilibrium2013 muscle model, which maps directly onto MuJoCo’s muscle model. Second, we converted the OpenSim model to MuJoCo using MyoConverter, a validated pipeline tested across multiple musculoskeletal systems. MyoConverter performs (i) XML conversion of geometry and attachment points, (ii) muscle kinematics alignment to match moment arms, and (iii) muscle kinetics alignment to match force–length curves. We applied all steps and verified that conversion errors were minimal; we now include the full MyoConverter report in the revised manuscript.
>
> To further address transferability, we repeated our muscle synergy analysis using the MuJoCo-generated muscle activations. The resulting synergies closely matched those obtained in OpenSim (except in the stiffness + armature condition), and remained antagonistic and non-overlapping. These results are now presented in Supplementary Figure 7.
> Finally, we note that our aim is not to identify a single “true” muscle activation profile, but to explore a range of modeling and control configurations that produce muscle dynamics plausibly aligned with biological behavior.
>
> 3. We agree that the plots alone do not explain how well the reconstructed joint movements are. To address this, we include videos of the optimized model alongside ground-truth movements in the Supplementary Material.
>
> **Questions:**
> 1. We apologize for the confusion. In Figure 3C, the label should have been unitless instead of arbitrary units (AU), as we plot the RMSE of joint angles (deg) normalized with respect to the joint angle range (deg). We have fixed it in the revised manuscript. In Figure 4A, we used Arbitrary Units because the reward term is a collection of values from different units.
>
> 2. We agree that showing the reference is helpful. We have updated Figure 4A to overlay the reference joint angles (dashed lines) against the simulated trajectories, allowing for a direct visual comparison.
>
> 3. The stiffness reference angles were taken from the resting pose of the fly. The coefficients for stiffness, damping, and armature were empirically selected to maintain a stable posture (i.e., eliminating unphysical jitter) in the absence of muscle activation. Precise identification of these passive parameters remains a challenge in the field due to the difficulty of manipulating forces in such small, living organisms.
>
> We hope that our answers will address the Reviewer’s concerns.

---

### Official Review · Reviewer_E3vf · 2025-11-01

**Soundness:** 2
**Presentation:** 4
**Contribution:** 2
**Rating:** 6
**Confidence:** 4

**Summary:**

This paper discusses the addition of muscles to the NeuroMechFly drosophila body model. 15 muscle-tendon units were added to each of the forelegs, based on micro-CT imaging. The authors used Hill-type muscle models, and tuned the parameters to reconstruct captured tethered walking and grooming data. They then used imitation learning to learn policies to perform walking and grooming behavior. They also analyzed muscle synergies and examined the effect of passive forces on the policy learning process.

**Strengths:**

1. The paper contributes a new component to a whole-body model of the fruit fly: anatomically realistic leg muscles.
2. The authors integrate anatomical imaging data, behavior data, muscle and joint dynamics modeling, and imitation learning in an interesting way. Their pipeline seems like a reasonable starting point that others can learn from, especially with respect to selecting which muscles parameters to optimize and the actual process of optimizing them.
3. The paper is well-written and the results are presented clearly.

**Weaknesses:**

1. The model the authors present only has muscles in parts of the front two legs, and the work doesn't deal with ground contact or external forces, which could make it difficult to directly apply when modeling many behaviors.
2. It's a little hard to tell without frame-by-frame comparisons, but it seems like the reconstruction quality presented in panel 3c might not be that high, at least for walking data.
3. I'm not sure what to make of the muscle synergy analysis. If a meaningful set of synergies was really discovered, I would expect to see a knee in panel 4c, but it doesn't seem like there is one. The explained variance numbers are a little hard to interpret as well; it might be more interesting to know how the fit quality would degrade using 1-, 2-, or 3-dimensional control vectors (with learned per-muscle weights) instead of controlling each muscle independently.
4. The motivating claim on line 451, "Placing a model of the musculoskeletal system—an additional layer of processing—between the policy network’s output and physical actions stabilizes the control task by making the action space better formed and more error-tolerant", does not seem to be substantiated. To support this claim, I think the authors would need to compare policy learning in models with and without muscles and per-joint passive forces.
5. Unless I missed it, there isn't a direct comparison to the original version of the NeuroMechFly model the authors are extending. This would probably help contextualize the behavior fitting results.

**Questions:**

1. What exactly are the inputs to the static optimization procedure used to fit muscle parameters? Model structure + instantaneous joint positions, velocities, and accelerations? And what is the loss function?
2. Why does panel 3c use arbitrary units? Either degrees or (unitless) joint range fractions would be easier to interpret.
3. The plot labels in panel 3d might be a little confusing to some readers. I think I eventually figured out the shorthand, but it took me a minute.
4. On line 381, what is meant by "systematic evaluation"? Do the curves plotted correspond to the best models after discovered via a grid search, in which case shouldn't the red-orange family be expected to have an advantage, since there's a larger space to explore?
5. Where did the reference angles for the joints come from? Would it be worth attempting to discover better values for these?
6. In terms of reproducibility, it looks like the tracked keypoint locations are included in the supplementary material, which should be helpful to readers interested in replicating the authors' results. Are the behavior videos and/or the imaging data available as well, or will they be made available?

---

> ### Author Response · Authors · 2025-11-27
>
> We thank the Reviewer for their positive assessment of our work.
>
> **Response:**
>
> 1. We agree that this is an important point. We initially focused on the forelegs because high-quality anatomical data were available and because forelegs play central roles in well-studied Drosophila behaviors, making them a natural starting point. The middle and hind legs are included in our model, and we plan to refine their muscle routes as additional anatomical data become available.
>
> In the revised manuscript, we now enable body–environment contacts during locomotion. Because the middle and hind leg muscle models are not yet fully optimized, these legs (and the right foreleg) were controlled with position controllers, while muscle activations from the trained policy drove the left foreleg. Remarkably, without any additional tuning, the muscle-driven foreleg generated sufficient forces to maintain a stable stance and support ground contact.
>
> We have added a supplementary video and ground-reaction-force plots illustrating this contact-enabled locomotion. Ongoing work focuses on optimizing the middle and hind leg muscles to achieve fully muscle-driven multi-leg locomotion.
>
> 2. We agree that static plots cannot fully convey the quality of the motion tracking. To address this, we have uploaded a video comparing the optimized model simulations alongside the ground-truth experimental data in the Supplementary Material (Video 1). We also refer the Reviewer to the Supplementary Figure 4 for a trajectory-based comparison.
>
> 3. Thank you for this great suggestion. Following your recommendation, we reconstructed muscle activation profiles using increasing numbers of synergies and ran forward-dynamics simulations for each reconstruction. Although one or two synergies explained over 80% of the variance in the muscle activity data, they were insufficient to reproduce the ground-truth joint kinematics with high fidelity. In contrast, using three synergies yielded qualitatively more accurate joint-angle reconstructions. We additionally quantified reconstruction accuracy using the root-mean-square error and found that grooming kinematics could be recapitulated with fewer synergies than locomotion, suggesting that grooming can be controlled by lower dimensions. These analyses have been added to the revised manuscript as a new Appendix section (A6) and Supplementary Figure S6, and were referred to in the main text.
>
> 4. We agree with the Reviewer that this claim was not explicitly tested by comparing torque-driven vs. muscle-driven learning in this specific manuscript. As this is tangential to our primary results regarding passive properties and model reconstruction, we have removed this sentence from the revised manuscript to avoid unsupported claims in the discussion.
>
> 5. We do compare the accuracy of the reconstructed kinematics in the updated biomechanical model with the original Neuromechfly in Supplementary Figure 3C, along with the changes in the biomechanics (Supplementary Figure 3A-B). We are happy to include additional comparisons if the Reviewer has other suggestions.
>
> **Questions:**
>
> 1. We used the standard OpenSim Static Optimization tool. The inputs are the target kinematics (joint angles) at each time step t, and the muscle model & parameters. The objective function minimizes the sum of squared muscle activations at each time step, subject to the constraint that the muscle torques must equal the inverse dynamic torques required to track the motion. We will modify the diagram in Figure 3B to further clarify this.
>
> 2. We apologize for the confusion. The y-axis in Figure 3C represents the RMSE normalized by the joint's range of motion (unitless). We chose to represent the data this way because a nominal value of RMSE alone is not meaningful without considering how much the DOF moves. We have updated the figure label to be precise.
>
> 3. We agree with the Reviewer. We replaced the abbreviations with their full versions in the revised manuscript.
>
> 4. By "systematic evaluation," we refer to testing of every combination of passive properties (stiffness, damping, armature) as shown in Figure 5B. It does not imply a grid search for optimal values. The coefficients were empirically selected to ensure a stable resting posture with no muscle activations, since a precise system identification is not yet technically feasible in Drosophila.
>
> 5. The reference joint angles were calculated from high-resolution 3D pose estimation of behaving flies using DeepLabCut and Anipose. We validated these angles by verifying that the forward kinematics of the model accurately reproduced the 3D keypoint positions (Figure S3C). Therefore, we believe that the joint angles used in this study are accurate.
>
> 6. We could not include the behavioral recordings in our submission due to size limitations. Upon publication, we will make all video recordings, trained model checkpoints, and code publicly available.

---

### Official Review · Reviewer_TPoP · 2025-11-03

**Soundness:** 4
**Presentation:** 4
**Contribution:** 4
**Rating:** 8
**Confidence:** 5

**Summary:**

This paper presents a 3D musculoskeletal model of drosophila legs. They propose a data-driven approach to building these models and unknown muscle parameters are estimated by an optimization approach, quite similar to system identification techniques. Once these detailed muscle models have been built, the authors present results on muscle activation patterns, muscle synergies and the effect of passive joint properties like damping, stiffness on muscle activates using Reinforcement Learning technique to mimic recorded drosophila leg movements for two distinct tasks - locomotion and grooming.

**Strengths:**

The paper is very well written, structured and easy to read. The level of detail, extensive experiments and the thoroughness of the evaluation presented in this paper needs to be appreciated. The problem statement is well motivated and its relation to existing research clearly presented.

The visualizations used in the paper, both of the muscles as well as quantitative results, are great and are really helpful for a reader to understand the text better.

The results presented in the paper are very insightful and come from well thought out experiments. The discussions about muscle synergies and effect of passive joint properties will definitely inspire future research.

**Weaknesses:**

While the results are promising, I found that certain details make the descriptions in the paper confusing. For example, All through section 3.2 and 3.3, one of the tasks being described are locomotion and most readers will have a mental image of a drosophila walking. But, in the limitations and future work section, it’s mentioned that body-body and body-environment contact forces are ignored - does this mean that the trajectory following was done without any contact between the leg and a surface? Adding to the confusion - In the appendix - the authors mention “We concentrated our efforts on developing a fully functional front-leg muscle model for two main reasons”. Does this mean that only the front two leg motion were simulated in the experiments for locomotion? Clearly describing the exact task would be very helpful to the reader.

Some details about imitation learning would be a great addition to the paper - what was the dimensionality of the state and action space?

**Questions:**

For most muscles in figure S6, the learned activations seem to be either 0 or 1 across all passive joint properties. Can the authors comment on the plausibility of this in a real musculoskeletal system?

Typically for reward functions in reinforcement learning for locomotion or any in general imitation of a recorded trajectory - there is a term that penalizes effort. For example, sometimes this can be minimizing joint torques, accelerations, muscle activations, etc.. I'm curious if this was considered during experiments?

Im also interested in the authors thoughts about the relationship between muscle synergies and passive joint properties, would it make sense to experiment how the synergies changes as joint properties change?

Comment - Typically exploration in reinforcement learning is challenging, especially when the action space is muscle activation. There has been some work in addressing this - https://openreview.net/forum?id=C-xa_D3oTj6 . The idea is not to naively explore but to identify some state dependent exploration that helps learning efficiency. This could be a useful tool for next set of experiments using reinforcement learning.

---

> ### Author Response · Authors · 2025-11-27
>
> We thank the Reviewer for their positive assessment of our work.
>
> **Response:**
> The Reviewer is correct that the model reproduces the walking kinematics of the front legs without enabling the body-environment contacts where the model was suspended in the air. We agree that those points were not clearly articulated in the main text, and we will make the necessary adjustments to the manuscript.
>
> Upon requests from other reviewers, in the revised manuscript, **we now enable full contact interactions between the fly and the environment for locomotion.** Because our middle and hindleg muscle reconstructions are not yet fully optimized, we controlled those legs (and the right foreleg) using position controllers, while actuating the left foreleg with the muscle activations produced by the trained policy from the armature & damping & stiffness model. Remarkably, without any additional training, the muscle-driven left foreleg generated sufficient forces to maintain stable stance and support ground contact.
>
> We have added a new supplementary video demonstrating ground locomotion with contact enabled, along with the corresponding ground-reaction forces for all six legs (five PD-controlled and one muscle-driven). We are currently working on optimizing the muscle model parameters for the middle and hind legs to achieve a fully muscle-driven multi-leg locomotion.
> Finally, our decision to fully model the front legs was made for several reasons: (i) high-quality anatomical datasets were available, and forelegs play key roles in complex, well-studied behaviors–making them a natural starting point for neuromechanical modeling.
>
> We apologize for the omission. The action space is a 15-dimensional vector representing the activation levels $[0, 1]$ of all muscles in a single leg. The observation space (86 dimensions total) consists of:
> Cartesian coordinates of four key points (coxa-femur, femur-tibia, tibia-tarsus, and leg tip) relative to the body ($3 \times 4 = 12$ dims).
>
> Joint angles and velocities for the 7 DoFs ($7 + 7 = 14$ dims).
> Muscle states: activations, forces, lengths, and velocities for all 15 muscles ($4 \times 15 = 60$ dims).
> We have added these details to the Methods section in the revised paper.
>
> **Questions:**
> 1. Thank you for this observation. Indeed, several muscles in Figure S6 (new Figure S7) exhibit almost binary activation patterns. Although we cannot claim that this precisely reflects biological activation amplitudes due to the lack of data on fly physiology, such patterns are plausible in insects.
>
> In particular, the fast fibers in Dropsphila, which were modeled in this work, are known to operate in rapid, high-force bursts rather than smooth graded activations. These muscles are often recruited phasically to generate the quick, high-impulse forces characteristic of insect limb movements.
>
> Thus, although the RL controller may exaggerate the discretization of activation, the overall burst-like structure is consistent with known insect neuromuscular physiology.
>
> 2. We did not explicitly include an effort penalty (e.g., minimizing squared muscle activation) in the reward function. We instead relied on the passive joint properties to constrain the motion naturally. The armature and damping coefficients inherently resist high-frequency jitter and excessive velocity, while the stiffness (spring) assists in returning the limb to a resting pose. We speculate that these physical constraints regularized the motion without needing an explicit auxiliary penalty term. Comparing explicit effort penalties against passive property constraints is an interesting direction for future work.
>
> 3. We conducted a muscle synergy analysis for each passive joint configuration, and the results are now included in Supplementary Figure 7 of the revised manuscript. With the exception of the armature + stiffness condition, the synergies extracted from MuJoCo muscle activations qualitatively resembled those obtained from OpenSim. In particular, we consistently observed the same two antagonistic synergies at the beginning of the cycle.
>
> Some differences across conditions are expected, as OpenSim’s Static Optimization minimizes instantaneous muscle effort to match a target pose, whereas our RL framework optimizes a cumulative reward dominated by tracking accuracy. These differing objectives can lead to distinct muscle coordination patterns.
>
> 4. We thank the Reviewer for this suggestion. In this work, we focused on quantifying learning efficiency across different biomechanical configurations rather than optimizing the RL algorithm itself. However, we agree that state-dependent exploration is relevant. We have added a reference to DEP-RL in our Discussion section.

---

### Official Review · Reviewer_eZKz · 2025-11-03

**Soundness:** 3
**Presentation:** 3
**Contribution:** 2
**Rating:** 4
**Confidence:** 4

**Summary:**

The authors of this manuscript developed a pipeline to estimate the properties of leg muscles for simulation directly from X-ray scans. They then applied them to imitation learning experiments to make predictions about muscle synergies employed across different behaviors, and investigate the influence of passive joint properties on learning.

**Strengths:**

The muscles in musculoskeletal modeling are a much-needed part of understanding the neural control of movement. This has been sorely lacking in biomechanical simulations of Drosophila, but this paper aims to address that gap.

1. The details of their optimization-based approach to estimating muscle parameters is interesting, since experimentally estimating these properties is challenging.

2. While the results about muscle synergies being different across behaviors are not surprising, the investigation of passive joint properties effect on learning is a clever utilization of MuJoCo joint parameters to address a scientific question about the advantage of using a combination of passive mechanics and active control.

**Weaknesses:**

1. The result that distinct synergies are employed across behaviors (walking vs. grooming) aligns with well-known concepts. While this confirms the model’s biological plausibility, it does not yield surprising scientific insight.

2. Environmental contact forces and interactions are excluded, restricting the ecological validity of locomotion simulations.

3. Although the optimization framework is solid, the lack of direct experimental validation (e.g., comparison with EMG) limits confidence in the accuracy of the predicted activations and synergies.

4. The topic is very niche.

**Questions:**

See weaknesses above.

---

> ### Author Response · Authors · 2025-11-27
>
> We thank the Reviewer for their positive assessment of our work. We believe that our model lays the foundation for further research using insect-scale musculoskeletal models in physics simulations.
>
> **Response:**
> 1. While the concept of muscle synergies is indeed well-established in motor control, the specific predictions regarding Drosophila leg muscle recruitment patterns remain unknown due to the lack of experimental data on individual muscle activity during behavior. Our work bridges this gap. By combining high-resolution anatomical datasets with behavioral kinematics, our model predicts which specific muscles are co-activated and when. For instance, we identify that the sternal rotator muscles (Sar, Sa) appear task-invariant, whereas the coxal muscles show task-specific specialization (Figure 4E).
>
> Beyond this, we demonstrate that our approach (i.e., actuating a biomechanically accurate body model to replicate experimental data) can be a viable and generalizable method to identify muscles comprising specific synergies. These are novel, testable biological hypotheses that cannot be derived from the connectome or behavior alone. We believe this predictive capacity is a significant scientific contribution.
>
> 2. Thank you for raising this important point. In the revised manuscript, **we now enable full contact interactions between the fly and the environment for locomotion** (also see Supp. Video 3). Because our middle and hindleg muscle reconstructions are not yet fully optimized, we controlled those legs (and the right foreleg) using position controllers, while actuating the left foreleg with the muscle activations produced by the trained policy from the armature & damping & stiffness model. Remarkably, without any additional training, the muscle-driven left foreleg generated sufficient forces to maintain stable stance and support ground contact.
> We have added a new supplementary video demonstrating ground locomotion with contact enabled, along with the corresponding ground-reaction forces for all six legs (five PD-controlled and one muscle-driven). We are currently working on optimizing the muscle model parameters for the middle and hind legs to achieve a fully muscle-driven multi-leg locomotion.
>
> 3. We acknowledge that direct validation via EMG would be ideal. However, **recording individual leg muscle activity in freely moving Drosophila is currently not feasible due to the extreme technical challenges associated with the animal's size and the lack of genetic driver lines targeting single leg muscles.** While some studies have recorded flight muscles (Melis et al, 2024) or femur muscles (using calcium imaging in restricted preparations) (Azevedo et al., 2020), these datasets do not contain the complex coxal and trochanteral muscles modeled in our work.
>
> Consequently, our optimization-based approach serves as the best currently available method to infer these internal states, providing a testbed to generate hypotheses that may guide the development of future experimental tools.
>
> 4. We respectfully disagree. Drosophila melanogaster is **one of the most popular model organisms in neuroscience. In particular, it is currently the only limbed animal with a fully mapped nervous system** (Dorkenwald et al., 2024; Phelps et al., 2021). This makes Drosophila an ideal model to bridge the gap between animal biomechanics and neural network dynamics at the interface between neuroscience and embodied machine learning. Indeed, the field is rapidly growing and moving toward "simulation neuroscience" –integrating connectomics with biomechanics to understand behavior (e.g., Vaxenburg et al., 2025; Wang-Chen et al., 2024; Lobato-Rios et al., 2022).
>
> Our work provides the missing musculoskeletal layer required to link neural circuits to physical action. **Given the broad interest in embodied AI, neuromechanics, and connectomics, we believe this work is timely and relevant to a wide audience at ICLR interested in biologically grounded motor control.**
>
> We hope that our answers will address the Reviewer’s concerns.

---

### Meta-Review · Area_Chair_5sxy · 2026-01-07

**Summary:**

The article presents a musculoskeletal simulation of movement in drosophila and analyses muscle synergies and passive forces.

Reviewers are generally positive, praising the level of detail, extensive experiments, and provided insights. Points of criticism are no surprising scientific insight, limitations of the simulations, lack of direct experimental validation, some of which have been discussed or addressed in the rebuttal.

I find the strengths outweigh the weaknesses. Therefore I recommend accept.

**Reviewer Concerns:**

Reviewer eZKz:

does not yield surprising scientific insight.
-> Response in rebuttal

Environmental contact forces and interactions are excluded
-> Addressed in rebuttal

lack of direct experimental validation
-> Response in rebuttal

**Reviewer Scores:**

For each review, specify how you think the reviewer would have changed their score if they had been able to participate fully in the discussion.

Reviewer eZKz: 4 -> 6
Reviewer TPoP: 8 -> 8
Reviewer E3vf: 6 -> 6
Reviewer Lmjv: 6 -> 6

---

### Decision · Program_Chairs · 2026-01-26

Accept (Poster)